# Alternating Projections With Volume Sampling

## Abstract

The method of Alternating Projections (AP) is a fundamental iterative technique with applications to problems in machine learning, optimization and signal processing. Examples include the Gauss-Seidel algorithm which is used to solve large-scale regression problems and the Kaczmarz and projections onto convex sets (POCS) algorithms that are fundamental to iterative reconstruction. Progress has been made with regards to the questions of efficiency and rate of convergence in the randomized setting of the AP method. Here, we extend these results with volume sampling to block (batch) sizes greater than 1 and provide explicit formulas that relate the convergence rate bounds to the spectrum of the underlying system. These results, together with a trace formula and associated volume sampling, prove that convergence rates monotonically improve with larger block sizes, a feature that can not be guaranteed in general with uniform sampling (e.g., in SGD).

## 1 Introduction

Solving a system of linear equations is one of the most fundamental computational problems. It is core to many problems in optimization, machine learning and signal processing that require solutions to large-scale linear systems. Advances in computational efficiency of solving large-scale linear systems directly translate to more efficient algorithms for non-linear optimization (Wilson et al., 2021). Randomization of classical algorithms such as coordinate descent and gradient descent has been instrumental in solving large-scale optimization problems. Characterizing the convergence of these randomized algorithms for solving linear systems is key to optimizing their performance and developing acceleration techniques that are used for more general classes (e.g., strongly convex) of objective functions. Broadly speaking, these randomized descent algorithms are row-space and column-space methods for solving a consistent linear system $Ax = b$, that in the randomized setting can be viewed as a sequence of alternating projections onto certain subspaces (see Section 2) specified by $A$. When a single column/row is selected at each iterate (e.g., coordinate descent or SGD) these are rank-1 projections onto hyperplanes, and more generally when a subset of columns/rows are chosen (e.g., block coordinate descent with size $n$) the resulting rank-$n$ projections are onto subspaces with codimension $n$.

Due to their sequential stochastic nature, these algorithms lend themselves to a Markovian view that facilitates the use of ergodic theory of (continuous state space) Markov chains to determine convergence criteria as well as rates of convergence. Analyzed as a time-homogeneous Markov chain, the existence and uniqueness of the stationary measure concentrated at the solution of the linear system is guaranteed by a stochastic version of the contraction mapping theorem. Likewise, the rate of convergence can be bounded using the *spectral gap* of a certain operator (see Section 2) that is constructed, for each algorithm, from $A$. While these results are well understood in Markov chain theory, establishing the relationship between the spectral gap—and hence the rate—to the spectrum of $A$ has been a fundamental theoretical challenge. Understanding this relationship is of practical importance since performance of competing iterative methods such as Krylov subspace methods (e.g., conjugate gradients) are well understood in terms of the spectrum of $A$ (Saad, 2003).

For the base case of rank-1 projections (e.g., coordinate descent or SGD), when rows/columns of $A$ are sampled, i.i.d., with probabilities according to their lengths (Strohmer & Vershynin, 2009; Leventhal & Lewis, 2010), the spectral gap is simply determined from the smallest singular value

of $\boldsymbol{A}$. We demonstrate that for rank-$n$ projectors (e.g., block coordinate descent), the spectral gap is entirely determined from all of the singular values of $\boldsymbol{A}$ when subsets are sampled, i.i.d., according to their volumes. Establishing the relationship between the spectral gap and the singular values of $\boldsymbol{A}$ is a significant departure from standard results on (deterministic) block methods. In block methods the rate depends on the condition of the worst block in a partition of $\boldsymbol{A}$ – a quantity that depends on the performance of the partitioning method and is decoupled from the spectrum of $\boldsymbol{A}$.

Our theory shows the exact process by which the spectral gap improves with $n$. We demonstrate that the singular values of $\boldsymbol{A}$, or equivalently eigenvalues of $\boldsymbol{A}^T\boldsymbol{A}$ are nonlinearly transformed with an attraction towards their mean as $n$ increases. This evolution, we show, is the $n^{\text{th}}$ step in a recursive formulation of the Cayley-Hamilton theorem known as the Faddeev–LeVerrier algorithm.

To sample with probabilities according to volumes when $n$ is relatively small (as large as $n = 15$, as our experiments show) one can employ rejection sampling; for larger $n$ efficient volume sampling is made possible by more sophisticated techniques that were developed by the pioneering work of (Deshpande & Rademacher, 2010; Deshpande et al., 2006).

## 2 RANDOMIZED DESCENT ALGORITHMS

Randomized algorithms for solving a linear system of equations $\boldsymbol{A}\mathbf{x} = \mathbf{b}$, by and large, descend on an objective function that guarantees almost sure convergence. The specifics of the objective function as well as the descent varies from technique to technique. In this section we show that the method of alternating projections presents a unifying perspective for the analysis of these techniques.

Given an $N \times N$ positive definite matrix $\boldsymbol{A}$, the randomized Gauss-Seidel algorithm updates the iterate, $\mathbf{x}_k$, a single coordinate at a time which is chosen at random. The objective function being minimized here is $f(\mathbf{x}) = \|\boldsymbol{A}^{1/2}(\mathbf{x} - \mathbf{x}_\star)\|^2$, with $\mathbf{x}_\star$ being the solution to the linear system. A descent along a direction $\mathbf{d}$ with exact line search updates the iterate $\mathbf{x}_k$ according to (Leventhal & Lewis, 2010):

$$\mathbf{x}_{k+1} = \mathbf{x}_k + \frac{\langle \mathbf{d}, \mathbf{b} - \boldsymbol{A}\mathbf{x}_k \rangle}{\langle \mathbf{d}, \boldsymbol{A}\mathbf{d} \rangle}\mathbf{d} = \mathbf{x}_k + \mathbf{d}\left(\mathbf{d}^T\boldsymbol{A}\mathbf{d}\right)^{-1}\mathbf{d}^T(\mathbf{b} - \boldsymbol{A}\mathbf{x}_k). \tag{1}$$

Randomized Gauss-Seidel, chooses $\mathbf{d}$ to be a randomly-chosen coordinate vector $\mathbf{e}_n$ (i.e., all zeros except along the $n^{\text{th}}$ coordinate axis of $\mathbf{x}$) resulting in a coordinate descent in iterations $\mathbf{x}_k \to \mathbf{x}_\star$.

Similarly the randomized Kaczmarz algorithm considers the objective function $f(\mathbf{x}) = \|\mathbf{x} - \mathbf{x}_\star\|^2$ for a consistent system of equations. Given a matrix $\boldsymbol{A} \in \mathbb{R}^{M \times N}$ with $M \geq N$ rows, the descent direction $\mathbf{d}$ is chosen randomly from rows of $\boldsymbol{A}$. To facilitate accessing subsets of rows, we consider the collection of rows in $\boldsymbol{A}$ as a set and denote by $\boldsymbol{a} \in \boldsymbol{A}$ a vector whose transpose, $\boldsymbol{a}^T$, is some row of $\boldsymbol{A}$, and by $b_{\boldsymbol{a}}$ the corresponding element of $\mathbf{b} \in \mathbb{R}^M$ on the right hand side of the system $\boldsymbol{A}\mathbf{x} = \mathbf{b}$. The one-step optimal descent along this direction is accomplished by:

$$\mathbf{x}_{k+1} = \mathbf{x}_k + \frac{b_{\boldsymbol{a}} - \langle \boldsymbol{a}, \mathbf{x}_k \rangle}{\langle \boldsymbol{a}, \boldsymbol{a} \rangle}\boldsymbol{a}. \tag{2}$$

This results in a descent in iterations $\mathbf{x}_k \to \mathbf{x}_\star$ that lies at the intersection of all hyperplanes represented by rows of $\boldsymbol{A}$.

Common to these descent algorithms is a notion of projection onto a subspace described by the underlying linear system $\boldsymbol{A}\mathbf{x} = \mathbf{b}$. For Gauss-Seidel these subspaces are the ones spanned by the rows $\boldsymbol{a} \in \boldsymbol{A}^{1/2}$ (with the $n^{\text{th}}$ row corresponding to the choice of $\mathbf{e}_n$ and $\|\boldsymbol{a}\|^2 = \boldsymbol{A}_{n,n}$) which is well defined for a symmetric positive definite matrix. In contrast, in Kaczmarz, the subspaces are simply spanned by rows $\boldsymbol{a} \in \boldsymbol{A}$. Let $\boldsymbol{P}_1^{\boldsymbol{a}} := \boldsymbol{a}\,\boldsymbol{a}^T/\|\boldsymbol{a}\|^2$ denote the rank-1 orthogonal projector onto the space spanned by $\boldsymbol{a}$. The dynamics of these iterations can be analyzed by the contraction they introduce in each step to an error vector $\mathbf{z}_k$ by alternating projections:

$$\mathbf{z}_{k+1} = (\boldsymbol{I} - \boldsymbol{P}_1^{\boldsymbol{a}})\mathbf{z}_k. \tag{3}$$

For the Gauss-Seidel method the notion of error vector is a residual measure $\mathbf{z}_k := \boldsymbol{A}^{1/2}(\mathbf{x}_k - \mathbf{x}_\star)$ and in the Kaczmarz iterations the error is simply $\mathbf{z}_k := \mathbf{x}_k - \mathbf{x}_\star$. When subspace projections $\boldsymbol{P}_1^{\boldsymbol{a}}$ are chosen at random, as discussed in the next section, the randomized iterations introduce a contraction in the error vector $\mathbf{z}_k$. This contraction can be quantified by analyzing the expected norm of the error vector, $\mathbb{E}\|\mathbf{z}_k\|^2$, that can be used to bound the rate of convergence.

## 2.1 CONVERGENCE RATE

Existence and uniqueness of the invariant measure for the Markov chain—the point mass at the solution of the linear system $\boldsymbol{A}\mathbf{x} = \mathbf{b}$—can be demonstrated under very general conditions (Hairer, 2018) for these randomized alternating projection algorithms. It is easy to show that randomized alternating projections are Feller Markov processes, which upon the application of the Krylov-Bogoliubov theorem guarantees the existence of an invariant measure. Conversely, *deterministic contraction*, a property of orthogonal projections, guarantees the uniqueness of this invariant measure (see (Hairer, 2018) for details).

Given a probability distribution for randomly (and independently) choosing a coordinate $\mathbf{e}_n$ in Gauss-Seidel, or choosing a row $\boldsymbol{a} \in \boldsymbol{A}$ in Kaczmarz, the geometric rate of convergence is dependent on the *mean* of the projections $\boldsymbol{P}_1^{\boldsymbol{a}}$ that this distribution engenders. In particular, a lower bound on the rate of convergence is obtained from the spectral radius of the average contraction in (3): $\lambda_{\max}\mathbb{E}\left[\boldsymbol{I} - \boldsymbol{P}_1^{\boldsymbol{a}}\right] = \lambda_{\max}\left(\boldsymbol{I} - \mathbb{E}[\boldsymbol{P}_1^{\boldsymbol{a}}]\right) = 1 - \lambda_{\min}\left(\mathbb{E}[\boldsymbol{P}_1^{\boldsymbol{a}}]\right)$. The key quantity in this bound is the *spectral gap* (i.e., the smallest eigenvalue) of the operator representing the average of all the noted rank-1 projectors:

$$\tau_1 := \lambda_{\min}\left(\mathbb{E}[\boldsymbol{P}_1^{\boldsymbol{a}}]\right). \tag{4}$$

To elaborate, given a starting point $\mathbf{x}_0$, the randomized Gauss-Seidel's residual decays geometrically as: $\mathbb{E}\|\mathbf{r}_k\|^2 \leq (1 - \tau_1)^k \|\mathbf{r}_0\|^2$ and Kaczmarz's error as $\mathbb{E}\|\mathbf{x}_k - \mathbf{x}_\star\|^2 \leq (1 - \tau_1)^k \|\mathbf{x}_0 - \mathbf{x}_\star\|^2$.

A *fundamental* challenge is to relate this spectral gap, $\tau_1$, to quantities that can be measured and computed from $\boldsymbol{A}$. While in the deterministic realm of alternating projections this quantity depends on a series of subspace angles that are difficult to compute (Galántai, 2005; Deutsch, 1995; Nelson & Neumann, 1987), in the randomized regime, the spectral gap (and hence the rate) is simply determined from the smallest singular value of $\boldsymbol{A}$ if the probability distribution is set to select a projector $\boldsymbol{P}_1^{\boldsymbol{a}}$ with a probability proportional to length of $\boldsymbol{a}$ squared: $pr(\boldsymbol{a}) \sim \|\boldsymbol{a}\|^2$. These probabilities are simply the diagonal elements of $\boldsymbol{A}$ in Gauss-Seidel and lengths of rows of $\boldsymbol{A}$ in Kaczmarz. This was first observed for the randomized Kaczmarz algorithm in (Strohmer & Vershynin, 2009) where $\mathbb{E}[\boldsymbol{P}_1] = \boldsymbol{A}^T\boldsymbol{A}/\operatorname{Tr}\left(\boldsymbol{A}\boldsymbol{A}^T\right)$ and $\tau_1 = \lambda_{\min}(\boldsymbol{A}^T\boldsymbol{A})/\operatorname{Tr}\left(\boldsymbol{A}\boldsymbol{A}^T\right)$ and then leveraged in the randomized Gauss-Seidel (coordinate descent) case (Leventhal & Lewis, 2010) where $\mathbb{E}[\boldsymbol{P}_1] = \boldsymbol{A}/\operatorname{Tr}\boldsymbol{A}$ and $\tau_1 = \lambda_{\min}(\boldsymbol{A})/\operatorname{Tr}\boldsymbol{A}$.

## 2.2 VOLUME SAMPLING

To facilitate faster convergence in practical applications, there has been a long line of research investigating block methods for iterative solvers such as block Gauss-Seidel and block Kaczmarz (Saad, 2003; Elfving, 1980) that have also been explored in randomized settings (Needell & Tropp, 2014; Liu & Wright, 2016; Gower & Richtárik, 2015; Tu et al., 2017). The critical difficulty in block methods is that the convergence rate depends on the worst condition number among all blocks in a given partition of $\boldsymbol{A}$ – a quantity that is disconnected from the spectrum of $\boldsymbol{A}$.

Generalizing (1) to coordinate descent on a block of $n$ coordinates, a matrix $\boldsymbol{D}$ containing $n$ coordinate vectors replaces $\mathbf{d}$, giving:

$$\mathbf{x}_{k+1} = \mathbf{x}_k + \boldsymbol{D}\left(\boldsymbol{D}^T\boldsymbol{A}\boldsymbol{D}\right)^{-1}\boldsymbol{D}^T(\mathbf{b} - \boldsymbol{A}\mathbf{x}_k). \tag{5}$$

$\boldsymbol{D}^T\boldsymbol{A}\boldsymbol{D}$ selects a principal minor of $\boldsymbol{A}$ that is chosen according to the descent coordinates. The subspaces for projections in this case are spanned by subsets of size $n$ chosen from rows of $\boldsymbol{A}^{1/2}$, that we denote by $\boldsymbol{A}_n \subset \boldsymbol{A}^{1/2}$ with $\boldsymbol{A}_n\boldsymbol{A}_n^T = \boldsymbol{D}^T\boldsymbol{A}\boldsymbol{D}$ (for Kaczmarz $\boldsymbol{A}_n \subset \boldsymbol{A}$). The rank-$n$ projectors that introduce contraction in each step of (3) are given by $\boldsymbol{P}_n := \boldsymbol{A}_n^T\left(\boldsymbol{A}_n\boldsymbol{A}_n^T\right)^{-1}\boldsymbol{A}_n$. Then the convergence rate is similarly bounded by $1 - \tau_n$ with the spectral gap of the expected projector:

$$\tau_n := \lambda_{\min}\left(\mathbb{E}[\boldsymbol{P}_n]\right). \tag{6}$$

We establish that the spectral gap is determined from an evolution of singular values of $\boldsymbol{A}$ towards their mean when subsets are chosen with probabilities proportional to the square of the volumes they subtend. Specifically the expected projector $\mathbb{E}[\boldsymbol{P}_n] = \sum_{\boldsymbol{A}_n} pr(\boldsymbol{A}_n)\boldsymbol{P}_n$ is formed with probabilities according to their volumes: $pr(\boldsymbol{A}_n) \sim \det(\boldsymbol{A}_n\boldsymbol{A}_n^T)$ which is simply the determinant of the corresponding minor of $\boldsymbol{A}$. As we will see the normalization constant that is the sum of all squared

volumes $\mathrm{vol}_n := \sum_{\boldsymbol{A}_n} \det(\boldsymbol{A}_n \boldsymbol{A}_n^T)$, can be calculated efficiently for any $n$ using a trace formula despite the combinatorial nature of all subset of size $n$. This means that for small $n$ (in fact, as large as $n = 15$, as our experiments demonstrate) one can employ simple rejection sampling techniques for volume sampling and for large $n$ more sophisticated Markov chain sampling techniques (Deshpande & Rademacher, 2010; Deshpande et al., 2006) provide efficient volume sampling.

## 3  RESULTS

We demonstrate that under volume sampling the spectrum of $\mathbb{E}[\boldsymbol{P}_n]$ as $n$ increases evolves from the spectrum of $\mathbb{E}[\boldsymbol{P}_1] = \boldsymbol{A}/N$ towards its mean (see Fig. 1 for an example). We prove that this evolution is described recursively by the Faddeev-LeVerrier algorithm traditionally used for computing coefficients of the characteristic polynomial of $\boldsymbol{A}$. For the Gauss-Seidel method with a block size $n$, the spectrum of $\boldsymbol{A}$ is transformed according to the spectrum of the matrix $\boldsymbol{\Phi}_n$, with $\boldsymbol{\Phi}_1 := \boldsymbol{A}$, defined recursively for $n > 1$ as:

$$\boldsymbol{\Phi}_n = \boldsymbol{\Phi}_1 \left( \frac{\mathrm{Tr}\, \boldsymbol{\Phi}_{n-1}}{n-1} \mathbf{I} - \boldsymbol{\Phi}_{n-1} \right). \tag{7}$$

We show that $\mathbb{E}[\boldsymbol{P}_n] = \boldsymbol{\Phi}_n / \mathrm{Tr}\, \boldsymbol{\Phi}_n$ This shows that the spectral gap $\tau_n = \lambda_{\min}(\boldsymbol{\Phi}_n)/\mathrm{Tr}\, \boldsymbol{\Phi}_n$ is a polynomial of degree $n$ over the spectrum of $\boldsymbol{A}$. For $n = 1$ the results of (Leventhal & Lewis, 2010) follows. For $n = 2$ this implies the spectrum of $\boldsymbol{A}$ is transformed by the quadratic polynomial $\Phi_2(x) = (\mathrm{Tr}\, \boldsymbol{A})\, x - x^2$. Specifically the spectral gap $\tau_2 = \lambda_{\min}(\boldsymbol{\Phi}_2)/\mathrm{Tr}\, \boldsymbol{\Phi}_2$ is the smallest eigenvalue after this quadratic transformation and $\mathrm{Tr}\, \boldsymbol{\Phi}_2$ is the sum of those transformed eigenvalues. More generally the spectrum of $\boldsymbol{A}$ is transformed according to the degree-$n$ polynomial:

$$\Phi_n(x) := \sum_{p=1}^{n} (-1)^{p-1} \mathrm{vol}_{n-p} x^p, \tag{8}$$

where $\mathrm{vol}_n := \sum_{\boldsymbol{A}_n} \det \boldsymbol{A}_n \boldsymbol{A}_n^T$ is the sum of squared volumes of all subsets of size $n$ that constitutes the normalization factor discussed in Section 2.2. We show that for any $n$, the trace formula $\mathrm{vol}_n = \mathrm{Tr}\, \boldsymbol{\Phi}_n$ provides an efficient computation of the normalization factor that avoids the combinatorially large computation over all subset of size $n$. This process is the same for Kaczmarz except for the starting point: $\boldsymbol{\Phi}_1 := \boldsymbol{A}^T \boldsymbol{A}$ which coincides with (Strohmer & Vershynin, 2009) for $n = 1$ and provides a similar evolution of eigenvalues of $\boldsymbol{A}^T \boldsymbol{A}$ for block sizes $n > 1$.

The evolution of spectrum in (7) introduces an attraction to their mean as they progress towards equalization as $\boldsymbol{\Phi}_N = \boldsymbol{I}$ according to the Cayley-Hamilton theorem. To illustrate the nature of this evolution, as $n$ increases, Fig. 1 shows the process on two examples of spectra with a linear decay and an exponential decay.

## 4  PROJECTIONS WITH VOLUME SAMPLING

As discussed in Section 3 when subsets of $n$ rows are selected $\boldsymbol{A}_n \subset \boldsymbol{A}^{1/2}$ according to their volumes, the expected projector $\mathbb{E}[\boldsymbol{P}_n] = \boldsymbol{\Phi}_n / \mathrm{Tr}\, \boldsymbol{\Phi}_n$ evolves in a recursive fashion described by (7). In this section we prove this result by establishing a combinatorial analysis of the set of projectors corresponding to all subsets $\boldsymbol{A}_n$ together with a recursive expansion for rank-$n$ orthogonal projectors that to the best of our knowledge is new.

### 4.1  RECURSIVE EXPANSION OF PROJECTORS

**Lemma 1.** *Let $U$ denote the span of $n$ linearly independent vectors $\{\boldsymbol{a}_1, \ldots, \boldsymbol{a}_n\}$. For each $1 \leq s \leq n$, let $\mathbf{P}_1^s := \boldsymbol{a}_s \boldsymbol{a}_s^T / \|\boldsymbol{a}_s\|^2$ be the orthogonal projector into the subspace spanned by $\boldsymbol{a}_s$, and $\mathbf{P}_{n-1}^{\bar{s}}$ be the orthogonal projector into the subspace $U_{\bar{s}}$ of $U$, spanned by all but $\boldsymbol{a}_s$. Furthermore, let the angle between $\boldsymbol{a}_s$ and $U_{\bar{s}}$ be denoted by $\theta_s$. Then the orthogonal projector into $U$ has the expansion:*

$$\mathbf{P}_n = \sum_{s=1}^{n} \frac{1}{\sin^2 \theta_s} \mathbf{P}_1^s \left( \mathbf{I} - \mathbf{P}_{n-1}^{\bar{s}} \right). \tag{9}$$

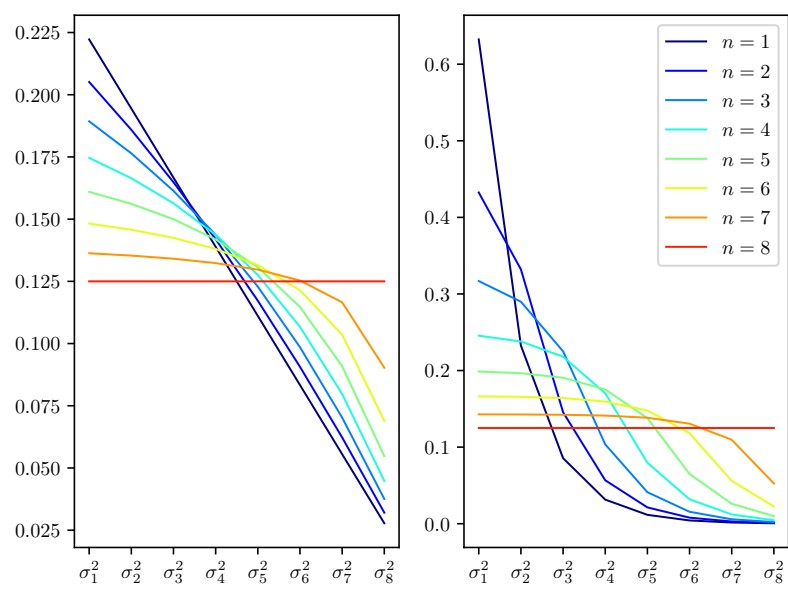

Figure 1: Evolution of spectrum towards its mean with exemplar linear and exponential decay with $N = 8$ according to volume sampling with (7).

*Proof.* Let $\mathbf{L}$ denote the expansion on the right hand side of (9). We prove $\mathbf{P}_n = \mathbf{L}$. First, we note that for any $t \neq s$, $\mathbf{P}_{n-1}^{\bar{s}} \boldsymbol{a}_t = \boldsymbol{a}_t$, which implies that $\left(\mathbf{I} - \mathbf{P}_{n-1}^{\bar{s}}\right) \boldsymbol{a}_t = \mathbf{0}$. We decompose $\boldsymbol{a}_s = \boldsymbol{a}_s^{\parallel} + \boldsymbol{a}_s^{\perp}$ into its components in the subspace $U_{\bar{s}}$ and orthogonal to that subspace: $\boldsymbol{a}_s^{\parallel} = \mathbf{P}_{n-1}^{\bar{s}} \boldsymbol{a}_s$ and $\boldsymbol{a}_s^{\perp} = \left(\mathbf{I} - \mathbf{P}_{n-1}^{\bar{s}}\right) \boldsymbol{a}_s$. The length of the orthogonal component is therefore $\|\boldsymbol{a}_s^{\perp}\| = \|\boldsymbol{a}_s\| \sin \theta_s$.

We now use these observations to show that

$$\mathbf{L}\boldsymbol{a}_t = \boldsymbol{a}_t \qquad 1 \leq t \leq n.$$

Based on the above, the only term in the summation corresponding to $\mathbf{L}\boldsymbol{a}_t$ that is non-zero is when $s = t$, so that

$$\mathbf{L}\boldsymbol{a}_t = \frac{1}{\sin^2 \theta_t} \mathbf{P}_1^t \boldsymbol{a}_t^{\perp} = \frac{\langle \boldsymbol{a}_t, \boldsymbol{a}_t^{\perp} \rangle}{\sin^2 \theta_t \|\boldsymbol{a}_t\|^2} \boldsymbol{a}_t.$$

Now $\langle \boldsymbol{a}_t, \boldsymbol{a}_t^{\perp} \rangle = \|\boldsymbol{a}_t\| \|\boldsymbol{a}_t^{\perp}\| \cos(\angle(\boldsymbol{a}_t^{\perp}, \boldsymbol{a}_t))$, and since $\angle(\boldsymbol{a}_t^{\perp}, \boldsymbol{a}_t)$ is complementary to $\theta_t = \angle(\boldsymbol{a}_t^{\parallel}, \boldsymbol{a}_t)$, we get $\langle \boldsymbol{a}_t, \boldsymbol{a}_t^{\perp} \rangle = \|\boldsymbol{a}_t\| \|\boldsymbol{a}_t^{\perp}\| \sin \theta_t$. Plugging in $\|\boldsymbol{a}_t^{\perp}\| = \|\boldsymbol{a}_t\| \sin \theta_t$ then gives the result.

For any vector $\mathbf{z}$ orthogonal to $U$, we have: $\mathbf{P}_{n-1}^{\bar{s}} \mathbf{z} = \mathbf{0}$ and $\mathbf{P}_1^s \mathbf{z} = \mathbf{0}$ since $\mathbf{z}$ is orthogonal to all the $n$ vectors $\boldsymbol{a}_1, \ldots, \boldsymbol{a}_n$; therefore, $\mathbf{L}\mathbf{z} = \mathbf{0}$.

This shows $\mathbf{L} = \mathbf{P}_n$ is the orthogonal projector into $U$, which is unique. $\qquad\square$

We expand on the notion of orthogonal projectors and introduce a quasi projector that is well-defined even when the vectors are not linearly independent:

$$\mathbf{Q}_n := \boldsymbol{A}_n^T \mathrm{adj}(\boldsymbol{G}_n) \boldsymbol{A}_n. \tag{10}$$

The adjugate matrix, $\mathrm{adj}(\boldsymbol{G}_n)$, is also the cofactor matrix of $\boldsymbol{G}_n := \boldsymbol{A}_n \boldsymbol{A}_n^T$ due to its symmetry. When the vectors are linearly dependent $\mathbf{Q}_n = \mathbf{0}$ (as we will see), otherwise the quasi projector is a scaled version of the orthogonal projector:

$$\mathbf{Q}_n = v_n^2 \mathbf{P}_n \quad \text{where} \quad v_n^2 := \det \boldsymbol{G}_n. \tag{11}$$

Here $v_n^2$ represents the (square of) $n$-volume of the parallelepiped formed by the rows chosen in $\boldsymbol{A}_n$. Using this volume definition, we can present a corollary to Theorem 1:

**Corollary 1** (Recursive Quasi Projector). *Under the assumptions of the lemma, let $v_1^s := \|\boldsymbol{a}_s\|$ and $v_{n-1}^{\bar{s}}$ denote the volume of the parallelepiped formed by all but $\boldsymbol{a}_s$. Moreover, let $\mathbf{Q}_1^s = \boldsymbol{a}_s\,\boldsymbol{a}_s^T$ and $\mathbf{Q}_{n-1}^{\bar{s}}$ denote the quasi projectors corresponding to $\mathbf{P}_1^s$ and $\mathbf{P}_{n-1}^{\bar{s}}$, respectively. Then we have:*

$$\mathbf{Q}_n = \sum_{s=1}^{n} \mathbf{Q}_1^s \left( \left( v_{n-1}^{\bar{s}} \right)^2 \mathbf{I} - \mathbf{Q}_{n-1}^{\bar{s}} \right). \tag{12}$$

*Proof.* The volume of the parallelepiped can be computed from the volume of any facet, $v_{n-1}^{\bar{s}}$, and the corresponding height, $v_1^s \sin \theta_s$: $v_n = v_{n-1}^{\bar{s}} v_1^s \sin \theta_s$. $\qquad\square$

**Lemma 2.** *For a linearly dependent set of vectors, $\{\boldsymbol{a}_1, \ldots, \boldsymbol{a}_n\}$, the quasi projector $\mathbf{Q}_n = \mathbf{0}$, the zero matrix.*

*Proof.* Based on (10), $\mathbf{Q}_n^T \mathbf{Q}_n = \mathbf{0}$ since $\boldsymbol{G}_n \text{adj}(\boldsymbol{G}_n) = (\det \boldsymbol{G}_n)\mathbf{I} = \mathbf{0}$. This implies that all columns of $\mathbf{Q}_n$ have zero norm. Hence $\mathbf{Q}_n = \mathbf{0}$. On the other hand, since Corollary 1 is under the linearly independent assumption, we show, using a continuity argument, that for a linearly dependent set Corollary 1 still holds.

The quasi projector, defined by (10) $\mathbf{Q}_n := \boldsymbol{A}_n^T \text{adj}(\boldsymbol{G}_n)\boldsymbol{A}_n$, has elements that are each a continuous function of vectors $\boldsymbol{a}_1, \ldots, \boldsymbol{a}_n$ that constitute rows of $\boldsymbol{A}_n$. This follows from the elements of the adjugate matrix (minors) being signed volumes of subsets of $\boldsymbol{a}$'s. Volume (determinant) is a continuous function of its set of vectors. Corollary 1 can therefore be expanded to include linearly dependent sets since any linearly dependent set of $n \leq N$ vectors in $\mathbb{R}^N$ can be perturbed to become a linearly independent set. $\qquad\square$

### 4.2 COMBINATORIAL ANALYSIS

We can now go back and perform an expectation analysis for $\mathbb{E}[\boldsymbol{P}_n]$ over all subset $\boldsymbol{A}_n$ of size $n$. As a reminder, for Gauss-Seidel with a symmetric positive definite $\boldsymbol{A} \in \mathbb{R}^{N \times N}$ there are the $\binom{N}{n}$ subsets of rows of $\boldsymbol{A}^{1/2}$ with each $\boldsymbol{A}_n \boldsymbol{A}_n^T$ being a principal minor of $\boldsymbol{A}$. In case of Kaczmarz with $\boldsymbol{A} \in \mathbb{R}^{M \times N}$ and $M \geq N$, there are $\binom{M}{n}$ subsets $\boldsymbol{A}_n$ that are simply the subsets of rows. The combinatorial arguments are identical for Gauss-Seidel and Kaczmarz and we present the more general argument based on $M$ rows that in the case of Gauss-Seidel simplifies to $M = N$.

We introduce a choice function that indexes these possible choices: $(i) \mapsto \{\boldsymbol{a}_1, \ldots, \boldsymbol{a}_n\}$ denotes the set of $n$ rows corresponding to the $i^{\text{th}}$ choice, $1 \leq i \leq \binom{M}{n}$. For example, $v_n(i)$ denotes the volume of the parallelepiped formed by the vectors from the rows selected for the $i^{\text{th}}$ choice, and likewise, $\mathbf{Q}_n(i)$ and $\mathbf{P}_n(i)$ denote the corresponding quasi projector and orthogonal projector.

We establish $\mathbb{E}[\boldsymbol{P}_n]$, as a polynomial in the matrix $\boldsymbol{A}$, when expectation is taken according to volume probability for the $i^{\text{th}}$ choice set proportional to $v_n^2(i)$. Recalling the definition of $\text{vol}_n$, sum of squared volumes $\text{vol}_n = \sum_i v_n^2(i)$, we have: $pr(i) = v_n^2(i)/\text{vol}_n$. Then the average projector is $\mathbb{E}[\boldsymbol{P}_n] = \sum_i pr(i)\mathbf{P}_n(i) = 1/\text{vol}_n \sum_i \mathbf{Q}_n(i)$. To state our key result, we define a total quasi projector for a matrix $\boldsymbol{A}$:

$$\boldsymbol{\Phi}_n := \sum_{i=1}^{\binom{M}{n}} \mathbf{Q}_n(i).$$

We now fully characterize $\boldsymbol{\Phi}_n$, specifically its spectrum, in our main result. When $n = 1$ the total quasi projector is $\boldsymbol{\Phi}_1 = \boldsymbol{A} = \sum_i \mathbf{Q}_1(i) = \sum_{\boldsymbol{a} \in \boldsymbol{A}^{1/2}} \boldsymbol{a}\,\boldsymbol{a}^T$ for Gauss-Seidel (and for Kaczmarz the Gram matrix: $\boldsymbol{\Phi}_1 = \boldsymbol{A}^T \boldsymbol{A} = \sum_i \mathbf{Q}_1(i) = \sum_{\boldsymbol{a} \in \boldsymbol{A}} \boldsymbol{a}\,\boldsymbol{a}^T$). For a larger set of rows $n > 1$ we show that $\boldsymbol{\Phi}_n$ is a degree-$n$ polynomial of the Gram matrix.

**Theorem 1** (Total Quasi Projector). *For $n > 1$ rows we have:*

$$\boldsymbol{\Phi}_n = \boldsymbol{\Phi}_1(\text{vol}_{n-1}\mathbf{I} - \boldsymbol{\Phi}_{n-1}) \tag{13}$$

*where $\text{vol}_n = \text{vol}_n(\boldsymbol{A}) := \sum_{i=1}^{\binom{M}{n}} v_n^2(i)$.*

*Proof.* Before arguing over the $\binom{M}{n}$ choices, we first expand the set of choices and define an operator that sums quasi projectors over all $n$-ordered choices of $M$ rows with replacement for a total of $M^n$ choices:

$$\tilde{\boldsymbol{\Phi}}_n := \sum_{j=1}^{M^n} \mathbf{Q}_n(j) = \sum_{\boldsymbol{a}_1 \in \boldsymbol{A}} \sum_{\boldsymbol{a}_2 \in \boldsymbol{A}} \cdots \sum_{\boldsymbol{a}_n \in \boldsymbol{A}} \mathbf{Q}_n.$$

We denote the sum of volumes in the expanded setting as $\tilde{\text{vol}}_n := \sum_{j=1}^{M^n} v_n^2(j)$. The expanded choices allow for summation over individual vectors that can sift through the recursion in (12). For example, the first term in (12) for $s = 1$ shows:

$$\sum_{\boldsymbol{a}_1 \in \boldsymbol{A}} \sum_{\boldsymbol{a}_2 \in \boldsymbol{A}} \cdots \sum_{\boldsymbol{a}_n \in \boldsymbol{A}} \mathbf{Q}_1^1 \left( (v_{n-1}^{\bar{1}})^2 \mathbf{I} - \mathbf{Q}_{n-1}^{\bar{1}} \right)$$

$$= \sum_{\boldsymbol{a}_1 \in \boldsymbol{A}} \mathbf{Q}_1^1 \sum_{\boldsymbol{a}_2, \ldots, \boldsymbol{a}_n \in \boldsymbol{A}} \left( (v_{n-1}^{\bar{1}})^2 \mathbf{I} - \mathbf{Q}_{n-1}^{\bar{1}} \right)$$

$$= \sum_{\boldsymbol{a} \in \boldsymbol{A}} \mathbf{Q}_1 \sum_{\boldsymbol{a}_1, \ldots, \boldsymbol{a}_{n-1} \in \boldsymbol{A}} \left( (v_{n-1})^2 \mathbf{I} - \mathbf{Q}_{n-1} \right)$$

$$= \boldsymbol{\Phi}_1 \left( \tilde{\text{vol}}_{n-1} \mathbf{I} - \tilde{\boldsymbol{\Phi}}_{n-1} \right).$$

Observing that the result of this summation is independent of $s$, allows us to establish:

$$\tilde{\boldsymbol{\Phi}}_n = n \boldsymbol{\Phi}_1 \left( \tilde{\text{vol}}_{n-1} \mathbf{I} - \tilde{\boldsymbol{\Phi}}_{n-1} \right).$$

Now we observe that in a particular choice of $n$-rows with replacement, if any row of $\boldsymbol{A}$ is selected more than once $v_n^2(j) = 0$ and $\mathbf{Q}_n = \mathbf{0}$ for any $n$. This means we can shrink the space of choices to $n$-permutations *without* replacement, with a total of $^M P_n := M!/(M-n)!$ choices, and still obtain the same $\tilde{\boldsymbol{\Phi}}_n$:

$$\sum_{j=1}^{^M P_n} \mathbf{Q}_n(j) = \tilde{\boldsymbol{\Phi}}_n = n \boldsymbol{\Phi}_1 \left( \tilde{\text{vol}}_{n-1} \mathbf{I} - \tilde{\boldsymbol{\Phi}}_{n-1} \right).$$

To further shrink the space of choices to $n$-combinations, we note that permuting the order of the rows in a particular choice does not change the squared volume of the parallelepiped they form. This means $\text{vol}_n = \tilde{\text{vol}}_n / n!$ for any $n$. Moreover, permuting the rows in a particular choice does not change the orthogonal projector $\mathbf{P}_n$ and consequently $\mathbf{Q}_n$. This means $\boldsymbol{\Phi}_n = \tilde{\boldsymbol{\Phi}}_n / n!$ for any $n$:

$$\boldsymbol{\Phi}_n = \frac{n}{n!} \boldsymbol{\Phi}_1 \left( (n-1)! \, \text{vol}_{n-1} \mathbf{I} - \tilde{\boldsymbol{\Phi}}_{n-1} \right) = \frac{1}{(n-1)!} \boldsymbol{\Phi}_1 ((n-1)! \, \text{vol}_{n-1} \mathbf{I} - (n-1)! \boldsymbol{\Phi}_{n-1})$$

$$= \boldsymbol{\Phi}_1 (\text{vol}_{n-1} \mathbf{I} - \boldsymbol{\Phi}_{n-1}).$$

$\square$

As a consequence, unwinding the recursion, we have:

**Corollary 2** (Polynomial Form)**.**

$$\boldsymbol{\Phi}_n = \sum_{p=1}^{n} (-1)^{p-1} \text{vol}_{n-p} \boldsymbol{\Phi}_1^p. \tag{14}$$

Since singular values of $\boldsymbol{A}$, when squared, match the eigenvalues of $\boldsymbol{\Phi}_1$, this establishes (8).

An important consequence of the theorem is that the volume measures, $\text{vol}_n$ associated with a set of vectors $\boldsymbol{a} \in \boldsymbol{A}$, can be recursively computed efficiently, which is of independent interest (McMullen, 1984; Dyer et al., 1998; Gover & Krikorian, 2010):

**Theorem 2** (Volume Computation of all $n$-subsets)**.** *Given a set of $M$ vectors, arranged in rows of $\boldsymbol{A}$, the sum of squared volumes of parallelepipeds formed by size-$n$ subsets is:*

$$\text{vol}_n = \sum_{i=1}^{\binom{M}{n}} v_n^2(i) = \frac{\text{Tr} \, \boldsymbol{\Phi}_n}{n}.$$

*The trace formula avoids the combinatorially-large computation over all subsets.*

*Proof.* Using the fact that $\mathrm{Tr}\,\mathbf{P}_n = \mathrm{rank}(\mathbf{P}_n) = n$, and linearity of trace, we have $\mathrm{Tr}\,\mathbf{Q}_n = nv_n^2$. Summing over all choices gives us:

$$\mathrm{Tr}\,\boldsymbol{\Phi}_n = \sum_{i=1}^{\binom{M}{n}} \mathrm{Tr}\,\mathbf{Q}_n(i) = n \sum_{i=1}^{\binom{M}{n}} v_n^2(i) = n\mathrm{vol}_n.$$

$\square$

This shows the recursion in (13) can be written entirely in terms of $\boldsymbol{\Phi}_1$:

$$\boldsymbol{\Phi}_n = \boldsymbol{\Phi}_1 \left( \frac{\mathrm{Tr}\,\boldsymbol{\Phi}_{n-1}}{n-1}\mathbf{I} - \boldsymbol{\Phi}_{n-1} \right).$$

This provides a recursive form of a Cayley-Hamilton expansion for $\boldsymbol{\Phi}_n$, in terms of powers of $\boldsymbol{A}$ for Gauss-Seidel (and powers of $\boldsymbol{A}^T\boldsymbol{A}$ for Kaczmarz) and their traces known as the Faddeev–LeVerrier algorithm.

## 5 EXPERIMENTS

We ran multiple instances of kernel ridge regression problems solved using randomized Gauss-Seidel for various values of block size $n$. We found that the convergence rates in each instance was patterned according to our theoretical predictions. We therefore present results from a prototypical numerical experiment.

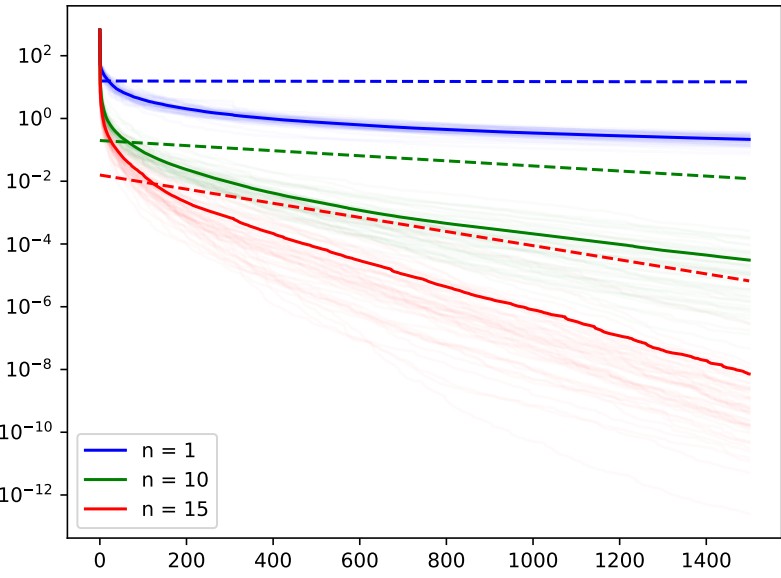

Figure 2: Convergence results for randomized Gauss-Seidel with block sizes of $n = 1$, 10, and 15. Dashed lines correspond to the rate bounds predicted by theory. The case $n = 1$ corresponds to the results of (Leventhal & Lewis, 2010). Each faint curve corresponds to a different run of the iterative algorithm. The solid lines correspond to ensemble averages across these trials. 30 trials were used to compute the ensemble average.

The details of the experimental setup was as follows. We generated multiple datasets $\{\langle \boldsymbol{a}_i, b_i \rangle\}_{i=1}^N$ where the dimensionality of $\boldsymbol{a}_i$ was set to be either 10, 25, or 100. Datasets ranged in size from $N = 25$ to 40. We limited the size of the dataset to a range for which a simple rejection sampler

could be used to sample rows according to their subtended volume. This was done to validate the theoretical results of the paper without the influence of any confounding effects introduced by a more sophisticated volume sampler (Deshpande & Rademacher, 2010) that while efficient, is influenced by burn-in and variance of stationary distribution effects. A shift invariant Gaussian kernel was then applied to the data to create one, two, or three clusters: $K_{ij} = \exp(-\gamma||\boldsymbol{a}_i - \boldsymbol{a}_j + C_{i,j}||_2^2)$, where $C_{i,j}$ was set to 0 if $\boldsymbol{a}_i$ and $\boldsymbol{a}_j$ belonged to the same cluster, and was set to randomly chosen and preset quantities for every pairing of disparate cluster memberships. This allowed us to experiment with varying levels of block dominance in $\boldsymbol{K}$ and concomitant effects on its spectrum. We then solved the linear system $(\mathbf{K} + \lambda\mathbf{I})\mathbf{x} = \mathbf{b}$ using randomized Gauss-Seidel where at each iteration $n = 1, 10$ or $15$ coordinates were simultaneously updated. The tuning parameters $\gamma, \lambda$ were set to various fixed values. Note that $(\mathbf{K} + \lambda\mathbf{I})$ is always symmetric positive definite for appropriately chosen $\lambda$ and therefore randomized Gauss-Seidel converges.

In each experiment 30 independent trials of randomized Gauss-Seidel were run using i.i.d. volume sampled rows of block size 1, 10 and 15. The modified residual $\mathbf{r}_k = (\mathbf{K} + \lambda\mathbf{I})^{1/2}(\mathbf{x}_k - \mathbf{x}_\star)$ was computed at each iteration and $\|\mathbf{r}_k\|^2$ was recorded. These residuals were then averaged across the trials to create an ensemble average, and was plotted in addition to the specific randomized trial of Gauss-Seidel. Also was plotted the rate bound as predicted by our theory. The results are presented in Figure 2.

The main implication of our results is establishing the explicit relationship between the rate of convergence in randomized AP (e.g., Gauss-Seidel) for solving $\boldsymbol{A}\mathbf{x} = \mathbf{b}$ to the spectrum of $\boldsymbol{A}$. As discussed in Section 2, $1 - \tau_n$ provides a bound on the rate of convergence and the spectral gap $\tau_n$ is simply the spectral gap of $\mathbb{E}[\boldsymbol{P}_n]$ that is related to the spectrum of $\boldsymbol{A}$ through (7). This recursive process stipulates the evolution of the spectrum of $\mathbb{E}[\boldsymbol{P}_n]$ as $n$ increases from 1 where the spectrum is that of $\mathbb{E}[\boldsymbol{P}_1] = \boldsymbol{A}/N$, towards its mean. We present an empirical validation of these theoretical results on the evolution of the spectrum of $\boldsymbol{A}$, and as a result show how the spectral gap $\tau_n$ increases as block size $n$ grows, in Figure 3. Note in particular that due to the recursive nature of the formula 8, the spectral gap can be computed very efficiently for problems of very large sizes. An appropriate choice of $n$ can then be made based on balancing the larger computational cost of volume sampling $n$ rows vis-a-vis the rate gains provided by a larger $n$.

# 6 CONCLUSION

This paper generalizes the results of (Strohmer & Vershynin, 2009; Leventhal & Lewis, 2010) that establish the relationship between the performacne of randomized Gauss-Seidel and Kaczmarz algorithms to the spectrum of $\boldsymbol{A}$ when single coordinates or rows (i.e., $n = 1$) are sampled according to their lengths. We establish that when $n > 1$ coordinates (or rows) are selected according to their volumes, the spectral gap that bounds the convergence rate is similarly determined from the spectrum of $\boldsymbol{A}$ that is nonlinearly transformed according to the Faddeev-LaVerrier algorithm. We derive a recursive formulation of this evolution of spectrum towards its mean and establish efficient volume computation results that avoids combinatorially large computations over subsets. These results establish the convergence of the method of alternating projections under volume sampling for which efficient algorithms have been developed in theoretical computer science.

# 7 REPRODUCIBILITY STATEMENT

The essential result of this paper is the evolution of the spectrum of a matrix $\boldsymbol{A}$ under the Faddeev-Laverrier algorithm is described recursively in (7) and in explicit polynomial form given in (8). Given the spectrum of $\boldsymbol{A}$ its evolution at *any* $n$ can be efficiently computed using the recursive $\boldsymbol{\Phi}_n$ and the volume computation formula $\text{vol}_n = \text{Tr}\,\boldsymbol{\Phi}_n$ that is established in Theorem 3. Hence the evolution process in Fig. 1 and Fig. 3 are reproducible. The convergence results in Fig. 2 are implementing (5) with randomness implemented by a simple rejection sampler. This experiment is also reproducible with minimal effort.

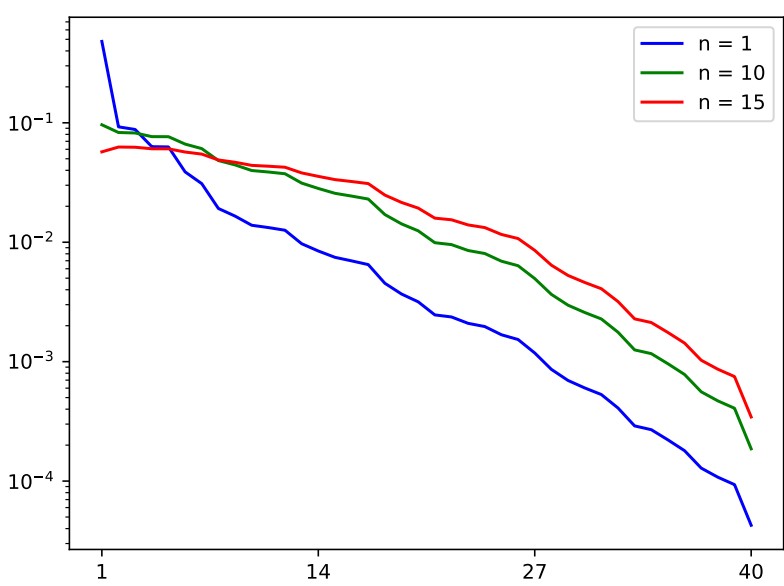

Figure 3: Log-scale view of spectrum of $\mathbb{E}[\boldsymbol{P}_1] = \boldsymbol{A}/N$ ($n = 1$), to $\mathbb{E}[\boldsymbol{P}_n]$ for $n = 10$ and $n = 15$ and the corresponding spectral gap, $\tau_n$, from the experiment in the previous figure and its evolution according to (7).

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
