# OpenReview forum: "Alternating Projections With Volume Sampling"
_ICLR.cc/2025/Conference — Submitted to ICLR 2025_

### Official Review · Reviewer_xhQm · 2024-10-28

**Soundness:** 2
**Presentation:** 3
**Contribution:** 1
**Rating:** 3
**Confidence:** 4

**Summary:**

This paper addresses the problem of solving the linear equation $ Ax = b $ using a randomized sequential approach, where in each iteration, the vector $ x $ is updated along a small number of randomly selected directions. Two specific instances of this approach, known as the Gauss-Seidel and Kaczmarz algorithms, are examined in detail. The convergence rate of this algorithm is characterized by the spectral gap of the associated Markov semigroup, which, in the context of this problem, is equal to the expectation of a certain random projector.

The paper's main contributions are presented in Theorems 2 and 3, where a recursive formula is derived for calculating this expectation. Specifically, if the randomly chosen blocks are of size $ n$, and the probability of selecting each block is proportional to the squared volume of the parallelepiped formed by the rows of  $A $ corresponding to that block, then the expectation $\Phi_n$ of the random projector satisfies the relation
$$
\Phi_n = \Phi_1 \left( \frac{{\mathrm{Tr}}(\Phi_{n-1})}{n-1} \mathbf{I} - \Phi_{n-1} \right).
$$
The paper provides full proof of this result and includes numerical experiments demonstrating the proposed methodology's practicality and effectiveness.

**Strengths:**

The paper deals with a simple problem that a large audience can understand.

The material is rather well presented.

**Weaknesses:**

1. In my view, the primary weakness of the paper lies in the relevance of its results to the ICLR community. While the problem addressed is indeed elegant in its simplicity, its broader applicability to machine learning remains unclear. Specifically, the linear equation $Ax = b$ can be viewed as finding the stationary points of the quadratic function $f(x) = \|Ax - b\|^2$. Consequently, a natural next step would be to explore how these results could extend to the more general problem of minimizing an arbitrary loss function $f(x)$.
  However, the paper lacks discussion on potential extensions in this direction. It would enhance the impact and relevance of the work if the authors demonstrated that the analyzed algorithms are, in fact, special cases of more general optimization algorithms applied to $f(x)$, with a quadratic function as a specific instance. Such an extension or contextualization would not only strengthen the theoretical insights but also significantly broaden the practical appeal to the ICLR audience, who are often focused on non-linear and complex loss landscapes in machine learning tasks.

2. I believe the presentation of the main results could be significantly enhanced. Specifically, the statements in Theorems 1 and 3 may not fully warrant their designation as theorems. Theorem 1 could be more appropriately presented as a proposition or lemma, while Theorem 3 might be better formatted as a numbered equation. In my view, the paper would benefit from having a single theorem that consolidates the claims currently in Theorems 2 and 3.

3. I have some doubts concerning the proof of Theorem 2. See the question in the next section.

4. Below are the typos that I found:

- line 74: I believe that the function $f(x)$ should be defined as $f(x) = \|A^{-1/2}(b - Ax)\|^2$, otherwise I do not manage to obtain Eq. (1).
- line 95: "rows of $a\in A$" -> "rows $a\in A$"
- line 108: "It is easy"
- line 131: the denominator $M$ should be replaced by the squared Forbenius norm of $A$, which is also the same as $\mathrm{Tr}(AA^T)$.
- line 132: the denominator $N$ should be replaced by the squared Forbenius norm of $A^{1/2}$, which is also the same as $\mathrm{Tr}(A)$.
- line 219: remove either "a" or "the" in "a the recursive"
- line 220: "analysis of the set"
- line 221: "subset $A_n$" -> "subsets $A_n$"
- Several places in Section 4.1: "projector to" -> "projector into"
- line 294: I do not understand the sentence ``We first establish $E[P_n]$ ..., and in the following section..."
- line 346: "volume squared" -> "squared volume" ?

5. There is an issue with how references are cited within the text. I recommend using the LaTeX command \citep when the reference is not integral to the sentence structure.

**Questions:**

The proof of Theorem 2 appears to contain flaws in its current form. While this issue might be possible to address, it does not seem to be a straightforward fix. Indeed, the proof relies on the fact that the right hand side of the equation on line 320 is equal to the expression given on line 326, with $1$ replaced by $s$ and summed over all $s$ from 1 to $n$. Here, the authors use Theorem 1. However, Theorem 1 is true when the vectors are linearly independent. The right-hand side (RHS) of the display on line 320 contains also $Q_n$s corresponding to linearly dependent $a_1,\ldots,a_n$. Of course, the corresponding term in the RHS of line 320 is then zero, but it is not clear to me that in such a case $\sum_{s=1}^n Q_1^s\big((v_{n-1}^{\bar s})^2\mathbf I - Q_{n-1}^{\bar s}\big) = 0$.

---

> ### Author Response · Authors · 2024-11-15
>
> We appreciate the time you have dedicated to examine the manuscript and the questions you raised. We are grateful for the attention to details and catching the typos.
>
> 1. **Relevant to ICLR Community?** Gauss-Seidel is the workhorse method for kernel methods appearing in classification and regression (e.g., Gaussian processes) that at the core require solving $A x = b$. We believe that the reviewer would agree that advances in the computational aspects of kernel ridge regression is of significant value to the ML community.  To wit, we list a sampling of articles, published in this field, that also echo the significance of these algorithms to ML:
> - Tu, Stephen, Shivaram Venkataraman, Ashia C. Wilson, Alex Gittens, Michael I. Jordan, and Benjamin Recht. "Breaking locality accelerates block Gauss-Seidel." In International Conference on Machine Learning, pp. 3482-3491. PMLR, 2017.
> - Hanzely, Filip, Nikita Doikov, Yurii Nesterov, and Peter Richtarik. "Stochastic subspace cubic Newton method." In International Conference on Machine Learning, pp. 4027-4038. PMLR, 2020.
>  - Wilson, Ashia C., Ben Recht, and Michael I. Jordan. "A Lyapunov analysis of accelerated methods in optimization." Journal of Machine Learning Research 22, no. 113 (2021): 1-34.
> - Défossez, Alexandre, and Francis Bach. "Averaged least-mean-squares: Bias-variance trade-offs and optimal sampling distributions." In Artificial Intelligence and Statistics, pp. 205-213. PMLR, 2015.
> - Jain, Prateek, Sham M. Kakade, Rahul Kidambi, Praneeth Netrapalli, and Aaron Sidford. "Parallelizing stochastic gradient descent for least squares regression: mini-batching, averaging, and model misspecification." Journal of machine learning research 18, no. 223 (2018): 1-42.
>
> 2. **General objective functions** As with other optimization methods, algorithms for solving $A x = b$ are often generalized to handle more general objective functions that belong to certain classes (see Défossez \& Bach in the above list). We have reasons to expect that our results maybe extended to strongly convex objective functions and the implications to larger class of objective functions is a subject of our current interest. However, we like to re-emphasize that linear solvers are ubiquitous and the key building block to nonlinear programming and optimization of more general objective functions (see for example Tu et al and Wilson et al in the above list).
>
> 3. **Given all the existing work, what's new here? Is this a significant contribution?** While stochastic gradient and coordinate descent methods are heavily studied in this context, we show the advantages of **volume sampling** that has never been explored before except for $n=1$. Larger batch sizes have been examined with uniform sampling before and it is well known that increasing batch sizes can not always increase convergence rate (e.g., Jain et al in the list above). We show that volume sampling guarantees that for larger batch sizes the convergence *always* improves. This result opens the door to sampling techniques to be used to design batch optimization algorithms that can guarantee improved performance for larger batch sizes. We respectfully submit that establishing volume sampling bounds for stochastic coordinate/gradient descent, along with guaranteed improvements, is a significant contribution.
>
> 4. **Proof of Theorem 2?** We point the reviewer to Corollary 1 (instead of Theorem 1) that works with quasi projectors that are well defined even for a linearly dependent set (see Lines 255-257). That corollary together with Lemma 1 addresses the question. As for presentation of results in lemmas or theorems, we agree with your suggestions. Please let us know if further clarification is necessary.
>
> Once again, we are grateful for your detailed review. Please let us know if you have further questions and if needed we would be happy to go to more details.

---

> > ### Comment · Reviewer_xhQm · 2024-11-18
> > **Proof of Theorem 2:**
> >
> > Thank you for your response. However, I am not entirely convinced by your explanation. Corollary 1 explicitly builds upon Theorem 1 and requires all the conditions of Theorem 1 to be satisfied. Since linear independence is one of these requirements, it follows that Corollary 1, as currently stated and proved, inherently depends on the assumption of linear independence.
> >
> > If the intent is for Corollary 1 to stand independently of this assumption, the statement and proof would need to be revised to explicitly address and resolve this discrepancy. As it stands, linear independence is an implicit yet unavoidable prerequisite for the validity of Corollary 1.

---

> > ### Comment · Reviewer_xhQm · 2024-11-18
> > **Relevant to ICLR Community**
> >
> > I have checked the first three papers from the list provided by the authors. These papers dedicate a portion of their content to the broader topic of minimizing a general function, extending beyond merely solving a linear equation. This observation reinforces my earlier point: to better align the work with the interests of the ICLR community, it is strongly recommended to clarify how the investigated framework connects to the general framework of loss function minimization.

---

> > > ### Author Response · Authors · 2024-11-18
> > >
> > > We agree with that observation and we would be happy to add the relevant discussion to the paper connecting the results to general optimization.
> > >
> > > However, we would like to re-emphasize the larger point that solving a system of linear equations is one the most fundamental computational problems. Improvements on computational aspects have direct impact in numerous applications in machine learning, signal processing and imaging. This paper contributes to the utility of fundamental ML algorithms like stochastic gradient/coordinate descent in improving computational solutions to this fundamental problem. These improvements would in turn advance the art in kernel methods by improving solvers.

---

> ### Author Response · Authors · 2024-11-18
>
> We appreciate the comment! Right after Theorem 1, on lines 254-255 we explicitly addressed this issue since the adjugate matrix exists even when determinant is 0.
>
> Based on your suggestion it make sense to be explicit about this in the statement of Corollary 1. Would that address your concern?

---

> > ### Comment · Reviewer_xhQm · 2024-11-19
> > **Flaw in the Proof of Theorem 2**
> >
> > No, it does not. What is written on lines 254-255 does not fix the issue. It merely implies that the left-hand side of Eq 12 is equal to zero (as mentioned in my original report). There is nothing in the paper that justifies the fact that in this case, the right-hand side of Eq 12 is also equal to zero. In its current form, the proof of Eq 12 is derived from Eq 9, which is proved only for a linearly independent family of vectors.

---

> > > ### Author Response · Authors · 2024-11-19
> > >
> > > We appreciate the nuanced argument you make! An argument that should address your concern is based on continuity of quasi projectors. We will add that to Lemma 1 that will then expand Corollary 1 to the case that you have highlighted.
> > >
> > > The quasi projector, defined by equation (10) $Q_n := A_n^T {\rm adj}(G_n) A_n$, has elements that are each a continuous function of vectors $a_1, \dots, a_n$ that constitute rows of $A_n$. This follows from the elements of the adjugate matrix (minors) being signed volumes of subsets of $a$'s. Volume (determinant) is a continuous function of its set of vectors. Corollary 1 can therefore be expanded to include *linearly dependent* sets since any linearly dependent set of $n \le N$ vectors in $\mathbb R^N$ can be perturbed to become a *linearly independent* set.

---

> > > > ### Comment · Reviewer_xhQm · 2024-12-02
> > > > **Note after the discussion**
> > > >
> > > > I would like to thank the authors for their prompt answers. My overall assessment of the paper did not change. The paper has interesting results, but the latter will benefit from a careful and substantial revision.

---

> > > > > ### Author Response · Authors · 2024-12-02
> > > > >
> > > > > Do you mind telling us what do you think needs to be substantially revised?
> > > > >
> > > > > You seem to have actually read the technical arguments in the paper (for which we are grateful), but the rationale behind your position is unclear to us. Perhaps your earlier comment "**merely solving a linear equation**" could be the reason.  We'd like to highlight that solving a linear system of equations is one of the most fundamental computational problems!
> > > > >
> > > > > An advance in solving linear systems would be a breakthrough with impact in many algorithms (optimization, ML). We are not claiming a breakthrough. However, we have found a new approach (with provable bounds) that shows that improvements in volume sampling would immediately translate to this problem. This suggests a previously-unexplored direction for solving this fundamental problem. We think our results provide an opportunity for MCMC experts to devise new algorithms that could lead to a breakthrough.

---

### Official Review · Reviewer_RfvD · 2024-10-30

**Soundness:** 3
**Presentation:** 3
**Contribution:** 2
**Rating:** 5
**Confidence:** 3

**Summary:**

This paper proposed a new algorithm for solving linear systems Ax=b, with adaptive minibatch sampling probabilities. The new approach incorporate volume sampling and tricks for reducing computational overhead to O(n) per iteration. The authors provide theoretical convergence analysis of the proposed method, demonstrating significant benefits over standard Gauss-Seidl methods. The numerical experiments validates the theoretical results.

**Strengths:**

The theoretical and algorithmic contribution appears to be solid and novel. To the best of the reviewer's knowledge this work is the first one to show such convergence rates is achievable. However, the reviewer is not familiar with this line of work, so could be overrating the novelty of the paper.

**Weaknesses:**

The numerical experiments seem to be very limited and preliminary. There is no baseline chosen for comparison. The reviewer believes that the authors should at least compare their new scheme with standard randomized block Gauss-Seidel. Real datasets should also be considered instead of synthetic data. See e.g. the experiment section of: Gower, R. M., & Richtárik, P. (2015). Randomized iterative methods for linear systems. SIAM Journal on Matrix Analysis and Applications, 36(4), 1660-1690.

It is unclear at the moment to the reviewer that, why this work is important enough to be published in ICLR -- as the practical benefits of the proposed approach over existing methods for Ax=b problem is not sufficiently demonstrated in the paper.

**Questions:**

Please improve the numerical studies, as mentioned above

---

> ### Author Response · Authors · 2024-11-19
>
> We appreciate the time and effort you have invested on our manuscript and the suggestions you made.
>
> - **Why this work is important enough?** Stochastic gradient and coordinate descent algorithms are widely used for large scale problems. For scaling computations, larger batch sizes are usually recommended; however, there is no guarantee that larger batch sizes lead to faster convergence. We have shown in this paper that volume sampling guarantees that. Now this opens the door to designing new algorithms based on volume sampling that can have a practical impact for large-scale problems (e.g., kernel ridge regression, Gaussian process classification/regression).
>
> - **Real datasets should also be considered.** Our theoretical result establishes the convergence bounds for randomized Gauss-Seidel for any problem $A x = b$, real or synthetic, large or small, and any batch size $n$ (Theorem 2). Experimental validation of this theory can only be presented when real rates, and eigenvalues, are computable so that we can observe how convergence bounds perform. The theory relates the convergence rates to eigenvalues of $A$ which can be practically computed in the ranges of experiments we have shown. We should also highlight that the pioneering work in this area (i.e., Strohmer & Vershynin, and Lewis & Leventhal), that established the rate at $n = 1$, have experiments of this nature. We are reporting the corresponding results for $n > 1$. The larger point here is the observation that rates improve, provably, for larger batch sizes, that these experiments corroborate.
>
> - **Baseline?** We would like to remind the reviewer that volume sampling bounds are only known for $n=1$ and our work shows the natural extension to $n > 1$ where there is no known convergence bounds (under volume sampling). Therefore, our base line in our experiments is $n=1$ and we show that larger batch sizes always improve rates.

---

### Official Review · Reviewer_ehie · 2024-10-31

**Soundness:** 2
**Presentation:** 3
**Contribution:** 2
**Rating:** 3
**Confidence:** 4

**Summary:**

The paper focuses on improving the efficiency and convergence rate of alternating projection methods in a randomized setting. By incorporating volume sampling for block sizes greater than 1, explicit formulas are derived to relate convergence rate bounds to the underlying system's spectrum. The authors argue that the results, combined with a trace formula and volume sampling, demonstrate that larger block sizes lead to monotonically improving convergence rates.

**Strengths:**

While the analytical investigations presented in the manuscript appear to be sound, their current impact and significance appear marginal.

**Weaknesses:**

This manuscript exhibits a disregard for advancements and established findings within the field (e.g., Chung et al. on ``Sampled limited memory methods for massive linear inverse problems'', Dereziński and Mahony on ``Recent and Upcoming Developments in Randomized Numerical Linear Algebra for Machine Learning''). Various authors have investigated block alternating projection methods which remains uncited. The authors' assertion that Stochastic Gradient Descent (SGD) is limited to one-row sampling is inaccurate. In fact, the SGD may include any number of samples of the matrix A.  The authors use a confusing non-standard mathematical notation employed throughout the manuscript making it harder for readers to follow. For instance, the conditional statement "a \in A" lacks clarity and rigorous definition. The authors' claim that the randomized Kaczmarz minimizes the objective function f(x) = ||x-x_\star||, where x_\star is the solution of Ax = b, makes no sense to me. Additionally, the authors do not make it accessible to the reader what algorithms are ``Feller'' means. The manuscript suffers from precision in methodology and analysis. Beyond that analytical investigations, and relevant applications are missing.

 A more comprehensive and in-depth analysis and a presentation of a large-scale application would be necessary to fully appreciate their potential contributions to the field.

**Questions:**

see above

---

> ### Author Response · Authors · 2024-11-16
>
> We thank the reviewer for the questions. Based on the nature of questions being asked here we would like to remind the referee about the central role that volume sampling plays in our paper.
>
> - **Assertion that Stochastic Gradient Descent (SGD) is limited to one-row sampling is inaccurate.** This is not what we claimed. Indeed SGD with various batch sizes allow for selecting any number of samples. Our assertion was: for *volume sampling* existing theory (i.e., convergence bounds) is limited to $n=1$ or one-row sampling due to Strohmer \& Vershynin for Kaczmarz and Lewis \& Leventhal for Gauss-Seidel. This is what we generalize in this paper.
>
> - **Objective function in Kaczmarz makes no sense.** The  $||x - x_*||^2$ can be counter-intuitive as one expects to see $A$  in the objective function. However, in each step, Kaczmarz performs an orthogonal projection -- minimizing $||x - x_*||^2$ along a row of $A$. The rows are specific descent directions (with optimal step size) for minimizing the said objective function. At its core, this is due to orthogonal projections providing minimizers of the $\ell_2$ distance.
>
> - **Impact and significance?** This is the first work that marries volume sampling to randomized (block) iterative methods. We show that volume sampling guarantees improvements in convergence as the batch size $n$ increases. This is a significant result since with uniform sampling it is well known that randomized iterative methods (e.g., gradient or coordinate descent) do not always benefit from larger batch sizes (see for example Jain, Prateek, Sham M. Kakade, Rahul Kidambi, Praneeth Netrapalli, and Aaron Sidford. "Parallelizing stochastic gradient descent for least squares regression: mini-batching, averaging, and model misspecification." Journal of machine learning research 18, no. 223 (2018): 1-42.). Our work opens a new possibility for designing randomized iterative methods, such as the Gauss-Seidel algorithm which is a workhorse for kernel methods in classification and regression (e.g., Gaussian process). Advances in computational aspects of kernel ridge regression is of significant value to the ML community.
>
> - **What about other block methods?** We re-emphasize that our work investigates the advantages of *volume sampling* by establishing new convergence bounds. Existing literature in block methods (including damped block methods) rely on uniform sampling of blocks (see for example the paragraph before eq 3 in Chung et al. you suggest). We have discussed and cited several pioneering works in block methods in Section 2.2. A fundamental difficulty with uniform sampling is that increasing block sizes does not always lead to faster convergence. Our theory demonstrates that with volume sampling this is guaranteed.
>
> - **Large-scale application?** The work presented here establishes the convergence bounds for randomized Gauss-Seidel for any problem $A x = b$, large or small, and any batch size $n$ (see Theorem 2). Experimental validation of this theory can only be presented when real rates  can be calculated so that we can observe how convergence bounds perform.
> - **Non-standard mathematical notation?** We employed the set-notation $a \in A$ to select a row of a matrix which is defined right at the beginning of section 2 lines 083-084. The advantage of this notation (compared to the index notation which is commonly used in literature on Kaczmarz algorithm) is that it eliminates the need for specifying indices since rows and subsets of rows are randomly drawn from $A$. We find this notation particularly suitable for block methods since we are concerned with all subsets of a given size from the set of rows in $A$.

---

> > ### Comment · Reviewer_ehie · 2024-11-24
> >
> > Thank you to the authors for their response. While they addressed most of my concerns, I remain unconvinced that efforts have been made to mitigate the issues I raised. For example, the authors note a difference in notation but have not provided any additional context or explanation for the reader.

---

> > > ### Author Response · Authors · 2024-11-25
> > >
> > > If all that remains is the matter of notation, we will be happy to add a paragraph describing why our notation simplifies the exposition. However, from the reviewer's comment it is not clear if there is any other issue that needs to be addressed (in addition to the ones we addressed previously).

---

> > > > ### Comment · Reviewer_ehie · 2024-11-26
> > > >
> > > > Thanks for the comment and no it is not just the notation. The manuscript maintains to suffers from precision in methodology and analysis. while relevant applications are missing.

---

> > > > > ### Author Response · Authors · 2024-11-26
> > > > >
> > > > > Thanks for the response.
> > > > >
> > > > > Your earlier comments suggested that notations were not defined, we misconstrued SGD and the objective function for Kaczmarz makes no sense. We believe we have responded to these concerns.
> > > > >
> > > > > Would you clarify what do you mean when you say that our work "**suffers from precision in methodology and analysis**"? We prove volume sampling convergence bounds using established theorems in linear algebra. If you see a flaw in the proof, then we could relate to your characterization, but currently we don't see how we could possibly excite you about our result.
> > > > >
> > > > > Now you can discount the significance of our results as being unimportant. We respectfully submit that solving large-scale system of equations is one of the most fundamental computational problems. Advances in this problem has impact on many problems in optimization and machine learning specifically (and scientific computing in general). We are not claiming we have a breakthrough, but our theoretical results show that a new approach is possible to attack this problem.  Our results presents an opportunity to MCMC experts who could contribute to a breakthrough in this fundamental problem.

---

### Official Review · Reviewer_7t6R · 2024-11-05

**Soundness:** 3
**Presentation:** 2
**Contribution:** 2
**Rating:** 5
**Confidence:** 3

**Summary:**

This paper study the spectral gap of the projection operators appearing in Randomized Gauss-Seidel and Kaczmarz method, and establish a bound that associates with the spectrum of the coefficient matrix in a recursive way.

**Strengths:**

The result of this paper is overall interesting.

**Weaknesses:**

While the result is interesting, whether it is useful or important is questionable, and it also seems not difficult to obtain the result. Moreover,  the organization and presentation of the paper can be improved. For example, the last line on pg. 1 is too long to understand; more details on the existing results of the randomized block Gauss-Seidel and randomized block Kaczmarz could be provided and compared so that the readers can appreciate the importance of the work more.

**Questions:**

1) Why $\lambda_\min$ is referred to as spectral "gap"?
2)  Assume $A\in\mathbb{R}^{m\times n}$ is a random Gaussian matrix. For randomized block, will it work well if the block size is $n$? If not, is there an interpretation based on the quantity provided in the paper?
3) Why only randomized block Gauss-Seidel is tested in the experiments?

---

> ### Author Response · Authors · 2024-11-18
>
> We appreciate referee's expression of interest and questions.
>
> - **Interesting, but utility?** Solving kernel methods such as kernel ridge regression (similarly in Gaussian process regression/classification) require solving large-scale inverse problems. The Gauss-Seidel (block/randomized) is widely used for solving practical problems (e.g., logistic regression on MNIST see Tu, Stephen, Shivaram Venkataraman, Ashia C. Wilson, Alex Gittens, Michael I. Jordan, and Benjamin Recht. "Breaking locality accelerates block Gauss-Seidel." In International Conference on Machine Learning, pp. 3482-3491. PMLR, 2017.) Advances in computational aspects of kernel methods is of significant value to the ML community.
>
>     Existing batch methods use uniform sampling of data to increase performance of Gauss-Seidel. However, it is well known that increasing batch size does not always improve overall performance/convergence rate (see for example Jain, Prateek, Sham M. Kakade, Rahul Kidambi, Praneeth Netrapalli, and Aaron Sidford. "Parallelizing stochastic gradient descent for least squares regression: mini-batching, averaging, and model misspecification." Journal of machine learning research 18, no. 223 (2018): 1-42.). Our results show that with volume sampling this improvement is always guarantees. If one has more computational cores (for parallel processing), increasing batch size under volume sampling always guarantees improvements.
> - **Why is $\lambda_{\min}$ is referred to spectral gap?** This terminology is commonly used (e.g., https://en.wikipedia.org/wiki/Spectral_gap) to show how far the system is to singularity (i.e., the gap is zero for singular system).
> - **Assume $A \in R^{M \times N}$ is a random Gaussian matrix. For randomized block, will it work well if the block size is $N$?** Interesting question! The answer for this question is Yes for *any* (full rank) matrix, not just random matrices! the spectral gap becomes largest possible which is $1/N$ (please see Figure 1 for an example where we change the number of rows (we call $n$) from 1 all the way to $N$).
> - **Why only randomized block Gauss-Seidel is tested in the experiments?** The focus in this paper is on randomized Gauss-Seidel as a fundamental difficulty with deterministic block methods is that the convergence rate is determined by the condition of the *worst* block. Designing partitioning/blocking algorithms that produces uniformly well-conditioned blocks from $A$ is a computationally-challenging problem when the matrix $A$ itself is not random. The randomization eliminates that problem which that explains its utility in large-scale machine learning, imaging and optimization problems.

---

> > ### Comment · Reviewer_7t6R · 2024-11-23
> >
> > Thank you for the reply.
> >
> > 1) While I understand the importance of solving large scale linear systems, I am still not sure whether volume sampling is very practical or not. Could you please  provide more details on the volume based sampling, e.g., the computational complexity of it, in addition to just mentioning several references?
> >
> > 2) I am not sure  whether "increasing batch size under volume sampling always guarantees improvements" is true or not in practice. For example, if $A\in\mathbb{R}^{m\times n} ~(m\ge n)$ is a Gaussian matrix,  by the random matrix theory, the condition number of any $n\times n$ block would be very large, it will even be unstable to solve this sub-system. Maybe more numerical experiments are desirable.
> >
> > 3) I didn't state the last question clearly. I was meaning why randomized block Kaczamarz method is not tested?

---

> > > ### Author Response · Authors · 2024-11-25
> > >
> > > Beautiful questions.
> > >
> > > 1. An advance in solving linear systems would be a breakthrough with impact in many fields. We are not claiming a breakthrough. However, we have found a new approach (with provable bounds) that shows that improvements in volume sampling would immediately translate to this problem. This suggests a previously-unexplored direction that could lead to a breakthrough.
> > >
> > >    Now let us tell you a bit more detail on volume sampling. To test our results, we implemented a vanilla rejection sampler: While going through subsets, we keep the value of the largest volume observed so far, and we reject a new volume based on its ratio to the max. For $1 \le n \le 15$, processing $n$ rows simultaneously in parallel (say with $n$ cores), in our experiments we used this rejection sampler to validate the bounds derived from this theory. Rejection sampling, as is well-known, is inefficient when the distribution peaks causing substantial rejections (e.g., for larger $n$). We are quite certain rejection sampling is not the optimal way to solve this problem. Sophisticated MCMC samplers such as the one described in Deshpande et al, are far more efficient. Their breakthrough was to approximate an exponential-time problem by a polynomial-time Markov Chain which is well recognized in theoretical computer science. They showed that their chain is geometrically ergodic and therefore it converges to the stationary distribution very fast (which is the distribution of volumes of subsets of size $n$ in our case).
> > >
> > >    We think our results present an opportunity to MCMC experts who could contribute to a breakthrough in solving linear systems.
> > >
> > > Coming to your second question, which is deep:
> > >
> > > 2. As you point out the condition number of subsets grow, but our framework introduces a preference for larger volumes over smaller ones which translates to a preference for lower condition numbers, rejecting high-conditioned subsets. Indeed when $n$ exceeds $spark(A)$ there are subsets with 0 volume (inf condition). These subsets are never visited according to the volume sampling. Avoiding such subsets efficiently (e.g., by MCMC) is at the core of volume-sampling bounds we have for solving $Ax = b$.
> > >
> > > 3. Our introduction points out that both Gauss-Seidel and Kaczmarz are instances of this more general framework of Alternating Projections (e.g., https://people.eecs.berkeley.edu/~brecht/papers/12.Recht.Re.amgm.pdf). The results for randomized Gauss-Seidel mirror those of Kaczmarz. In fact, our original results in computed tomography were on randomized block Kaczmaz. We considered kernel ridge regression more pertinent to the ICLR community and for space reasons we omitted the Kaczmarz results. We would be happy to re-include those in an appendix if the reviewer deems it necessary.

---

### Meta-Review · Area_Chair_a1jT · 2024-12-18

**Metareview:**

This work extends the Alternating Projections (AP) method to block sizes greater than one using volume sampling, providing explicit formulas linking convergence rates to the system's spectrum. It shows that convergence rates improve monotonically with larger block sizes, a guarantee not achievable with uniform sampling methods like SGD.

Although the paper's theoretical approach is interesting, the reviewers highlighted the following main weaknesses:

1. One reviewer questioned the practical applicability of the presented results.
2. Another pointed out the paper’s failure to acknowledge relevant advancements and established findings in the field.
3. A third reviewer noted that the numerical experiments are limited and preliminary, with no baselines selected for comparison.
4. Lastly, one reviewer questioned the paper’s relevance to the ICLR community.

**Additional Comments On Reviewer Discussion:**

All reviewers recommended rejection. The AC concurs with the reviewers' concerns and supports a recommendation for rejection. The AC also strongly encourages resubmission, recognizing the paper's importance and interest but acknowledging that it requires substantial revision.

---

### Decision · Program_Chairs · 2025-01-22

Reject